health and disease and epidemiology/ computational biology/ecology

computational biology, COVID-19, infectious diseases, epidemiology, time series

**Author for correspondence:**
Levente Kriston
e-mail: l.kriston@uke.de

# Assessing the strength of case growth trends in the coronavirus pandemic

## Levente Kriston

Department of Medical Psychology, University Medical Center Hamburg-Eppendorf, Martinistr. 52, 20246 Hamburg, Germany

LK, 0000-0003-0748-264X

The ability to distinguish between erratic and systematic patterns of change in case count data is crucial for assessing and projecting the course of disease outbreaks. Here, it is shown that measuring the strength of trends can provide information that is not readily captured by commonly used descriptive indicators. In combination with the 7-day moving average, Bandt and Pompe's permutation entropy and Wilder's relative strength index were found to support the timely detection of coronavirus epidemic trends and transitions in data from various countries. The results demonstrate that measuring the strength of epidemic growth trends in addition to their magnitude can significantly enhance disease surveillance.

## 1. Background

Governments, healthcare organizations and academic institutions routinely use daily case count data to monitor the spread of the new coronavirus pandemic. Mostly, they rely on averaging the number of new cases, calculating standardized incidence rates and estimating the rate of change over a specific period of time. For example, a study focusing on the rate of change in case count data up to the end of March 2020 found that daily growth rates should be permanently kept below 5% in order to avoid an exponential development of infections [1]. However, depending on the transmission paths, networks and dynamics behind the aggregated data, the identified patterns may or may not indicate lasting developments in the course of the pandemic [2–4].

Although simple descriptive indicators are valuable for assessing the magnitude of epidemic trends, they do not inform about the consistency or stability of the identified changes. However, distinguishing between erratic and systematic patterns of change is crucial for appraising and projecting the course of disease outbreaks. This point has not received much attention either in ecology or in epidemiology so far. Consequently, consensual methods for the measurement of the strength (consistency, reliability, stability, regularity) of perceived epidemic

trends are largely missing. In the present study, it was investigated whether indicators of trend strength developed in other fields of application offer insights about the course of the coronavirus pandemic beyond those provided by frequently used indicators of epidemic magnitude.

## 2. Methods

### 2.1. Data

Publicly available data on daily new case counts of infections was retrieved from the World Health Organization's Coronavirus Disease (COVID-19) Dashboard including reports up to 4 August 2020 [5] (electronic supplementary material, Data S1). Data on the population of countries were retrieved from the Department of Economic and Social Affairs of the United Nations on 4 August 2020 [6] (electronic supplementary material, Data S2).

In most countries, the first new coronavirus infections were detected in January or February 2020, thus, the end date for the analysis (4 August 2020) marks approximately the end of the initial half-year of the pandemic. Countries differed substantially regarding the development of their coronavirus case counts during this first half-year, providing an excellent opportunity to describe the behaviour of the proposed indicators in various settings. The present investigation was limited to this period with the assumption that the most common case count trajectory patterns had become visible by the beginning of August 2020 and the knowledge gain from extending the period of observation would have been marginal.

### 2.2. Measures

One indicator of trend magnitude and two indicators of trend strength were considered and interpreted using *a priori* defined thresholds. All calculations were performed in R [7] (Code S1).

In order to quantify the magnitude of epidemic trends, the 7-day moving average (MA) of standardized country-level daily new case count data was calculated for each day from the 6th day on after the first reported case in the respective country. Calculations were performed with the caTools package [8]. The values were standardized to express the daily incidence in 100 000 people. Consensual thresholds were not available for interpretation, therefore, guidelines were set in the present study. Based on a study on meningitis epidemics, a 7-day MA above 2.143 cases per 100 000 people (15 cases per 100 000 people in a week) was considered to indicate a considerable outbreak [9]. Following the guidance of the German government, an MA of 7.143 cases per 100 000 people (50 cases per 100 000 people in a week) was considered serious [10]. A twice as high average daily number of new cases (14.286 per 100 000 people, 100 per 100 000 per week) was categorized as critical.

The first measure of trend strength was Bandt and Pompe's permutation entropy, which originates from information theory and quantifies the complexity of a time series by calculating the entropy of the probability distribution of ordinal patterns [11]. Permutation entropy was calculated for each day form the 36th day on after the first case with an embedding dimension of 3 using MA data from the last 30 days including the index day using the statcomp package [12]. The embedding dimension of 3 was used because a comparably short-term indicator was preferred and 3 is recommended as the minimum by the original authors [11]. With this embedding dimension, a timeframe of 30 days is the lower limit of the number of necessary data points for calculations (from $5^*3! = 30$) [13]. In order to be able to deal with (highly unlikely) equal values in the time series, small random perturbations were added [11]. Here, permutation entropy was subtracted from 1 and considered to conceptualize predictability [14]. The permutation entropy-based predictability (PEBP) is based on the MA of the previous 30 days, ranges from 0 (complete irregularity) to 1 (complete regularity), and indicates only the presence but not the direction of trends. Generally applicable thresholds for interpretation are not meaningful for PEBP. Here, a Monte Carlo simulation was performed with 100 000 simulated time series, each containing 36 values drawn randomly from a normal distribution. For each time series, the 7-day MA (30 values in each series) and subsequently the PEBP were calculated as described above. The empirical distribution of PEBP indicated that 99% of the values generated from random Gaussian numbers were below 0.242, 99.9% below 0.326, and 99.99% below 0.391. Based on the assumption that values exceeding these thresholds are unlikely to be created by a completely erratic process, they were used to indicate possible, likely, and highly probable trends, respectively.

The second measure of trend strength was Wilder's relative strength index (RSI), which was developed for assessing the strength of a stock or a market by analysing daily closing prices over a

specific period of time [15]. The RSI is based on the MA of the previous 15 days, comprising information both on the strength and the direction of a trend and ranges from 0 (extreme downward trend) to 100 (extreme upward trend). The RSI was calculated for each day from the 21th day on after the first case from the 7-day MA data using the TTR package [16]. Daily change data for a 14-day period was used, as recommended by the original author [15]. Traditionally, the RSI is considered to suggest a downward trend below 30 and an upward trend above 70 [15]; therefore, these thresholds were used to indicate a possible trend. As using more strict thresholds is not unusual [17,18], values below 20 and above 80 were interpreted as likely and below 10 and above 90 as highly probable trends.

# 3. Results

## 3.1. Trend plots

The indicators were calculated and visualized for several countries which were considered to display various temporal patterns of their coronavirus case growth up to the beginning of August 2020. The number of daily new cases, the 7-day MA, the PEBP and the RSI are displayed together in trend plots with a common time axis in order to allow a comprehensive evaluation. In the following, the indicators for four countries showing stereotypical dynamics of the coronavirus epidemic are presented and discussed. These countries were chosen for illustration due to the particularly clearly recognizable patterns in their daily new case count data. Corresponding figures for countries similar to these stereotypes can be found in the electronic supplementary material.

## 3.2. Case studies

At the beginning of August 2020, some countries were experiencing a continuously growing epidemic wave. In India, the MA was constantly increasing and indicated an epidemic of considerable magnitude on 17 July 2020, the first time (figure 1). The PEBP exceeded the threshold for a possible trend on 18 March 2020 and signalled a likely and a highly probable trend on 23 March and 25 March 2020, respectively. Afterwards, it showed an increasing tendency until the end of the period of observation. The RSI suggested a highly probable upward trend on 4 March 2020 and remained above the threshold for a possible upward trend on all but one day. After 1 April 2020, it indicated a highly probable upward trend without interruption. A similar pattern was observable in Colombia (with the MA reaching a critical level; electronic supplementary material, figure S1), Mexico (with the strength indicators signalling a ceasing upward trend at the beginning of August; electronic supplementary material, figure S2) and Brazil (with partly conflicting strength indicators; electronic supplementary material, figure S3).

In the first half-year of the pandemic, case counts in several countries showed a clear epidemic wave that could be largely controlled with a low number of new cases for a longer period of time after the wave. In Germany, the MA exceeded the threshold for a considerable outbreak on 21 March 2020 and increased continuously until 5 April 2020, though always remaining below the threshold for a serious outbreak (figure 2). Subsequently, it decreased continuously and fell below the threshold for a considerable outbreak on 29 April 2020. After that it has been fluctuating on a low level. The PEBP indicated a possible, a likely, and a highly probable trend on 11 March, 14 March and 16 March 2020, respectively. A drop around the middle of April between two peaks signalled a trend change from accelerating to decelerating. On 11 June 2020, the PEBP fell below the threshold for a possible trend, indicated non-predictable new case counts for several weeks and signalled a trend roughly in the middle of July (a downward one, as shown by the MA). The RSI suggested a likely upward trend as early as 28 February 2020, remained very high until the first weeks of April and indicated a possible and a likely downward trend on 23 April and 30 April 2020, respectively. The RSI has exceeded the threshold for a possible and a likely upward trend again on 26 July and 30 July 2020, respectively, suggesting a possible second epidemic wave in Germany. Similar behaviour of the indicators can be observed in data from France (electronic supplementary material, figure S4), Italy (with the MA reaching a serious level at its peak; electronic supplementary material, figure S5) and Spain (with an apparent second wave building up during July; electronic supplementary material, figure S6).

Some countries experienced a second wave of infections after a longer period of control over the first outbreak. In Israel, the MA exceeded the threshold for an epidemic of considerable magnitude on 25 March 2020 and fell below this threshold again on 30 April 2020 without reaching a serious magnitude even at its

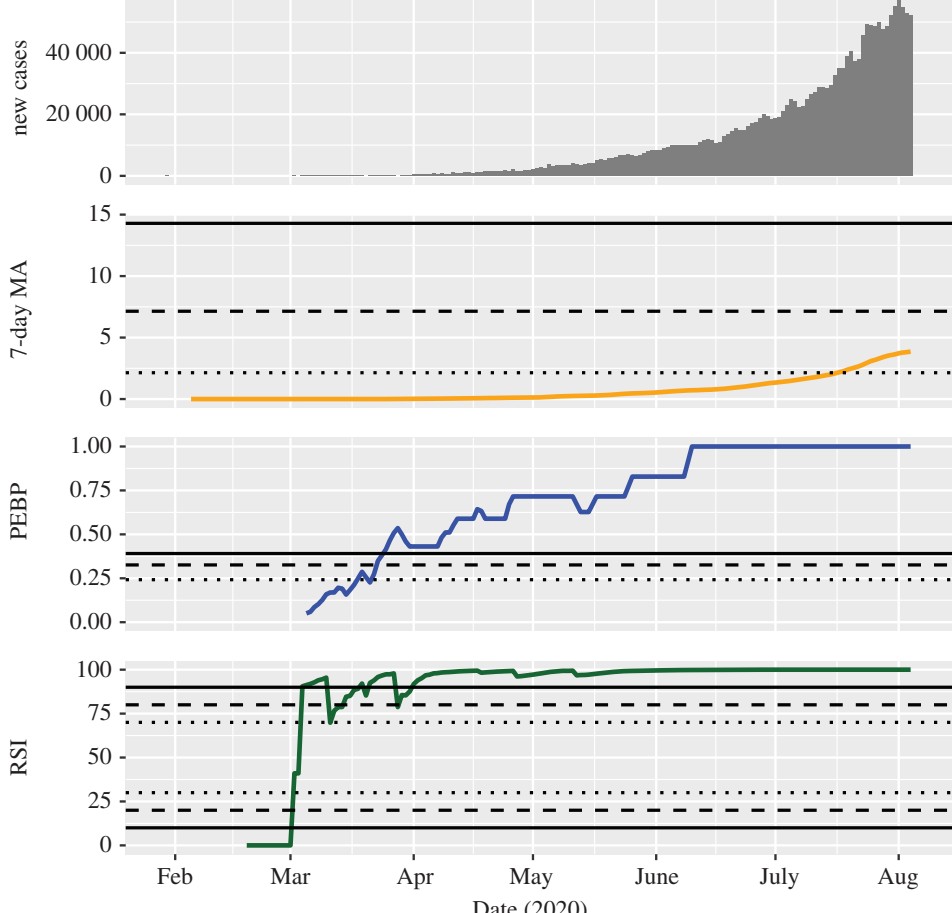

**Figure 1.** Development of the investigated indicators in India. For the 7-day moving average of the standardized incidence per 100 000 people (MA), dotted, dashed and solid lines indicate thresholds for considerable, serious and critical case counts, respectively. For the permutation entropy-based predictability (PEBP) and the relative strength index (RSI), dotted, dashed and solid lines indicate thresholds for possible, likely, and highly probable trends, respectively. As an example of the behaviour of the indicators in a continuously growing epidemic wave, it can be seen that they suggest rather early that a stable epidemic trend is present.

peak (figure 3). After a longer period of time, it began to increase again and crossed the threshold for a considerable, serious and critical number of infections on 18 June, 4 July and 13 July 2020, respectively. The PEBP could not be calculated before the first wave, but indicated a possible, a likely, and a highly probable trend on 23 June, 1 July and 2 July 2020, respectively. The RSI signalled a possible upward trend until 9 April 2020, suggested a possible downward trend from 3 May to 30 May 2020, and indicated a possible, a likely, and a highly probable upward trend on 11 June, 21 June and 28 June 2020. It fell below the threshold for a possible trend again on 1 August 2020. Further countries with a clearly identifiable second wave include Australia (electronic supplementary material, figure S7), Japan (with the number of infections constantly remaining below considerable; electronic supplementary material, figure S8) and Romania (electronic supplementary material, figure S9).

In some countries, acceleration of the first wave could be stopped but without reaching a substantial deceleration. In some of these cases, even a second wave hit upon the uncontrolled first one. In the United States, the MA exceeded the threshold for a considerable outbreak on 26 March and the threshold for the serious outbreak on 5 April 2020 (figure 4). It fell back below the serious level on 16 May 2020, but exceeded the threshold for a serious outbreak again on 21 June 2020. It reached even a critical level on 6 July 2020 and remained above this threshold after that. The PEBP indicated a possible and a likely trend on 30 March and 8 April 2020, respectively, fell below the threshold for a possible trend shortly thereafter and remained there for more than two months. It signalled a possible, a likely, and a highly probable trend on 1 July, 2 July and 3 July 2020, respectively. The RSI suggested the first possible upward trend on 21 February and remained consistently above this

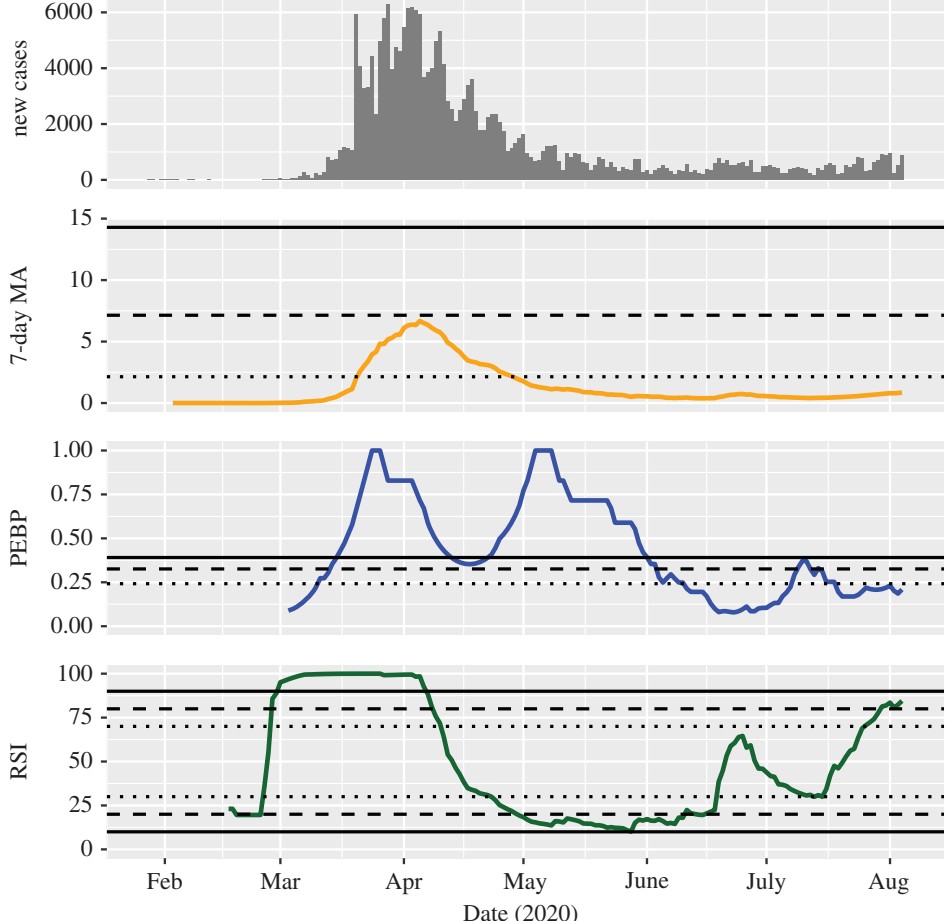

**Figure 2.** Development of the investigated indicators in Germany. For the 7-day moving average of the standardized incidence per 100 000 people (MA), dotted, dashed and solid lines indicate thresholds for considerable, serious and critical case counts, respectively. For the permutation entropy-based predictability (PEBP) and the relative strength index (RSI), dotted, dashed and solid lines indicate thresholds for possible, likely, and highly probable trends, respectively. As an example of the behaviour of the indicators in a clear epidemic wave that could be largely controlled with a low number of new cases for a longer period of time after the wave, it can be seen that they signal both the upward and the downward trends sensitively. Furthermore, it is apparent that they are reliable in indicating trend-free periods.

threshold after 4 March 2020, several times exceeding the threshold for a highly probable upward trend. It fell below the threshold for a possible upward trend at the end of April 2020 and signalled a possible, a likely, and a highly probable upward trend again on 23 June, 27 June and 3 July 2020, respectively. On 2 August 2020, it fell below the threshold for a possible upward trend again. Similar epidemic dynamics can be obtained in Iran (with a more controlled first and a less clear-cut second wave; electronic supplementary material, figure S10), Sweden (electronic supplementary material, figure S11) and Algeria (with an elongated first wave; electronic supplementary material, figure S12).

## 4. Discussion

The trend strength indicators provided information that was not clearly recognizable from the daily new case counts and their MA. First, they helped to confirm trends that otherwise were identified only by visual inspection of the MA. For instance, the continuously growing number of daily new infections in India was unequivocally categorized as a highly probable wave since the beginning of April (figure 1). Second, they were able to identify trend-free periods, like the weeks between the middle of June and the middle of July 2020 in Germany (figure 2). Third, they frequently indicated epidemic transitions before they became clearly visible in the MA of case counts, making them potential candidates for early warning signals of change [19]. For example, the critical second wave in Israel was anticipated with a high probability already at the very beginning of July 2020 (figure 3). Fourth,

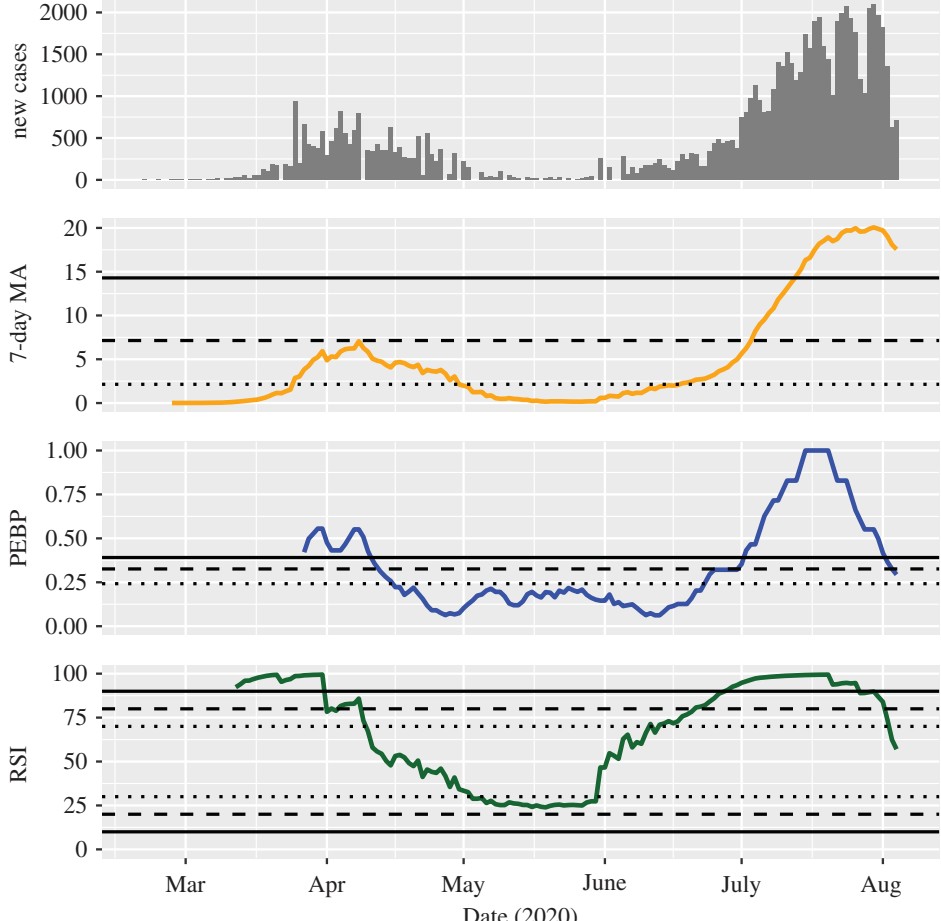

**Figure 3.** Development of the investigated indicators in Israel. For the 7-day moving average of the standardized incidence per 100 000 people (MA), dotted, dashed and solid lines indicate thresholds for considerable, serious and critical case counts, respectively. For the permutation entropy-based predictability (PEBP) and the relative strength index (RSI), dotted, dashed and solid lines indicate thresholds for possible, likely, and highly probable trends, respectively. As an example of the behaviour of the indicators in a second wave of infections after a longer period of control over the first outbreak, it can be seen that they warn very early that a new trend is manifesting itself.

they showed that decreasing numbers over a longer period of time do not necessarily indicate a downward trend, for instance in the months between the first and the second wave of infection in the United States (figure 4).

The present work began without a clear definition of key concepts like an epidemic wave or an upward trend. Considering that from a formal point of view even the most essential epidemiological terms are only vaguely and ambiguously defined [20–22], this might be attributed to the fact that infectious disease outbreaks are complex dynamical systems that are likely to elude unidimensional quantitative descriptions [23]. Based on the present study, however, an epidemic trend in a given region could be roughly defined as a specified pattern of change in a specified period of time in the past that is likely to continue largely unchanged for a specific period of time in the future with a given probability. For instance, a trend could be described as an exponential growth in the past 30 days with the number of daily new cases increasing by a factor of 1.05 (or 5%) every day that is expected to continue for the next two weeks by showing a less than 10% change in the logarithmic slope of the growth with a probability of 95%. The probabilistic statement at the end is what was termed the strength of a trend. It should be noticed that the investigated trend strength indicators signal only changes (e.g. trends with a substantial magnitude) directly and identify trends of no change (e.g. constantly low numbers of new infections) rather indirectly by not exceeding defined thresholds.

The results show that using more than one indicator of trend strength could be useful. In general, the PEBP is expected to react more slowly but more clearly to dynamical changes due to its ordinal data

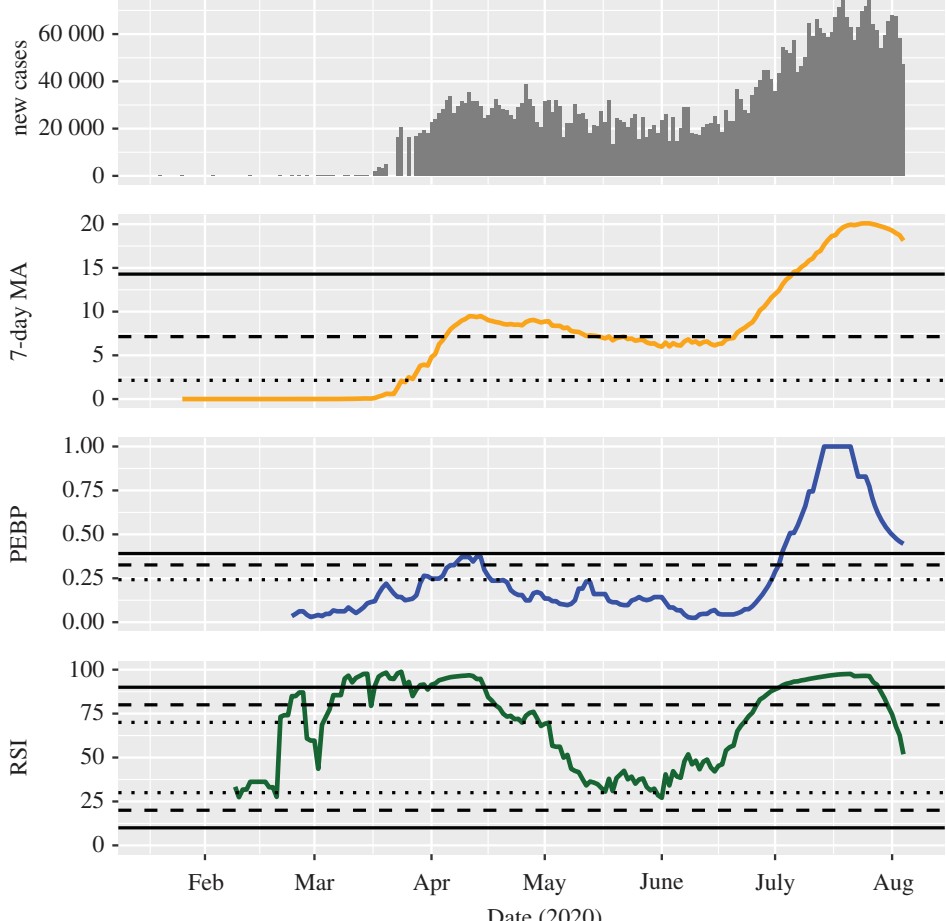

**Figure 4.** Development of the investigated indicators in the United States. For the 7-day moving average of the standardized incidence per 100 000 people (MA), dotted, dashed and solid lines indicate thresholds for considerable, serious and critical case counts, respectively. For the permutation entropy-based predictability (PEBP) and the relative strength index (RSI), dotted, dashed and solid lines indicate thresholds for possible, likely, and highly probable trends, respectively. As an example of the behaviour of the indicators when acceleration of the first wave could be stopped but without reaching a substantial deceleration, it can be seen that they are sufficiently specific and do not identify every decrease in case counts as a downward trend.

processing and longer timeframe. The RSI responds rather fast, but has the capacity to overreact to extreme irregularities in the data and produce false-positive results. Considering both of these strength indicators simultaneously and combining them with traditional measures of change patterns is most likely to lead to a more complete assessment of the dynamics of infectious disease outbreaks. Given that they are scalable, the indicators can be readily used to characterize trends with time horizons substantially exceeding the ones used here and to describe geographic regions and administrative territories of any size.

Data accessibility. Data and code are provided as electronic supplementary material.
Competing interests. We declare we have no competing interests.
Funding. The study was not externally funded.
Acknowledgements. The author is grateful to Dana Barthel for helpful comments on the manuscript.

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
