## [Reviewer comments · Royal Society Open Science]

Review History

RSOS-201622.R0 (Original submission)

Review form: Reviewer 1

Is the manuscript scientifically sound in its present form?

Yes

Are the interpretations and conclusions justified by the results?

Yes

Is the language acceptable?

Yes

Do you have any ethical concerns with this paper?

No

Have you any concerns about statistical analyses in this paper?

No

Recommendation?

Accept with minor revision (please list in comments)

Comments to the Author(s)

In "Assessing the strength of case growth trends in the coronavirus pandemic" authors study a critically relevant and timely problem, and they provide useful insights into how to better understand the spread of COVID-19 infections based on assessing the strength of case growth trends.

I have enjoyed reading, and I am in favor of publication after suitable revisions in the Royal Society Open Science.

A relevant recent reference to be added for a more detailed coverage of similar approaches is Forecasting COVID-19, *Front. Phys.* 8, 127 (2020) and Early spread of COVID-19 in Romania: Imported cases from Italy and human-to-human transmission networks, *R. Soc. Open Sci.* 7, 200780 (2020).

It would also improve the paper if the figure captions would be made more self contained. In addition to what is shown for which parameter values, one could also consider a sentence or two saying what is the main message of each figure.

Finally, it would be useful if the authors would make their source code available as supplementary material. This would promote the usage of the proposed method and allow others to take advantage of this research, and also allow them to reproduce the results.

Review form: Reviewer 2

Is the manuscript scientifically sound in its present form?

Yes

Are the interpretations and conclusions justified by the results?

No

Is the language acceptable?

Yes

Do you have any ethical concerns with this paper?

No

Have you any concerns about statistical analyses in this paper?

No

Recommendation?

Reject

Comments to the Author(s)

This is a nice exercise using a well-known method with data from the corona-virus pandemic. The selection of the time and the countries that were chosen is not clear: why India, Israel, Germany and the US? What is special about this countries? The time frame is too short until August. I am not convinced that moving average is adding substantially information to the many other models that have been published recently.

Decision letter (RSOS-201622.R0)

Dear Dr Kriston

The Editors assigned to your paper RSOS-201622 "Assessing the strength of case growth trends in the coronavirus pandemic" have now received comments from reviewers and would like you to revise the paper in accordance with the reviewer comments and any comments from the Editors. Please note this decision does not guarantee eventual acceptance.

Please submit your revised manuscript and required files (see below) no later than 21 days from today's (ie 29-Oct-2020) date. Note: the ScholarOne system will 'lock' if submission of the revision is attempted 21 or more days after the deadline. If you do not think you will be able to meet this deadline please contact the editorial office immediately.

on behalf of Professor Matjaz Perc (Associate Editor) and Pete Smith (Subject Editor)
openscience@royalsociety.org

Associate Editor Comments to Author (Professor Matjaz Perc):
Comments to the Author:

The two referees rated your work rather differently. We would be happy to consider a resubmission if you can make a strong case for the added value of your work to existing literature. Referee #2 expressed concerns in this regard.

Reviewer comments to Author:

Reviewer: 1

Comments to the Author(s)

In "Assessing the strength of case growth trends in the coronavirus pandemic" authors study a critically relevant and timely problem, and they provide useful insights into how to better understand the spread of COVID-19 infections based on assessing the strength of case growth trends.

I have enjoyed reading, and I am in favor of publication after suitable revisions in the Royal Society Open Science.

A relevant recent reference to be added for a more detailed coverage of similar approaches is Forecasting COVID-19, *Front. Phys.* 8, 127 (2020) and Early spread of COVID-19 in Romania: Imported cases from Italy and human-to-human transmission networks, *R. Soc. Open Sci.* 7, 200780 (2020).

It would also improve the paper if the figure captions would be made more self contained. In addition to what is shown for which parameter values, one could also consider a sentence or two saying what is the main message of each figure.

Finally, it would be useful if the authors would make their source code available as supplementary material. This would promote the usage of the proposed method and allow others to take advantage of this research, and also allow them to reproduce the results.

Reviewer: 2

Comments to the Author(s)

This is a nice exercise using a well-known method with data from the corona-virus pandemic. The selection of the time and the countries that were chosen is not clear: why India, Israel, Germany and the US? What is special about this countries? The time frame is too short until August. I am not convinced that moving average is adding substantially information to the many other models that have been published recently.

===PREPARING YOUR MANUSCRIPT===

===PREPARING YOUR REVISION IN SCHOLARONE===

-- If you have uploaded ESM files, please ensure you follow the guidance at <https://royalsociety.org/journals/authors/author-guidelines/#supplementary-material> to include a suitable title and informative caption. An example of appropriate titling and captioning

may be found at https://figshare.com/articles/Table_S2_from_Is_there_a_trade-off_between_peak_performance_and_performance_breadth_across_temperatures_for_aerobic_sc_ope_in_teleost_fishes_/3843624.

Author's Response to Decision Letter for (RSOS-201622.R0)

See Appendix A.

Decision letter (RSOS-201622.R1)

Dear Dr Kriston,

It is a pleasure to accept your manuscript entitled "Assessing the strength of case growth trends in the coronavirus pandemic" in its current form for publication in Royal Society Open Science. The comments from the Editors are included at the foot of this letter.

COVID-19 rapid publication process:

We are taking steps to expedite the publication of research relevant to the pandemic. If you wish, you can opt to have your paper published as soon as it is ready, rather than waiting for it to be published the scheduled Wednesday.

This means your paper will not be included in the weekly media round-up which the Society sends to journalists ahead of publication. However, it will still appear in the COVID-19 Publishing Collection which journalists will be directed to each week (<https://royalsocietypublishing.org/topic/special-collections/novel-coronavirus-outbreak>).

If you wish to have your paper considered for immediate publication, or to discuss further, please notify openscience_proofs@royalsociety.org and press@royalsociety.org when you respond to this email.

You can expect to receive a proof of your article in the near future. Please contact the production office (openscience_proofs@royalsociety.org) and the editorial office (openscience@royalsociety.org) to let us know if you are likely to be away from e-mail contact -- if you are going to be away, please nominate a co-author (if available) to manage the proofing process, and ensure they are copied into your email to the journal.

on behalf of Professor Matjaz Perc (Associate Editor) and Pete Smith (Subject Editor)
openscience@royalsociety.org

Associate Editor Comments to Author (Professor Matjaz Perc):

The author has much improved and extended his work, and responded to all comments constructively and comprehensively. My recommendation is therefore acceptance.

Appendix A

Response Letter

Associate Editor Comments to Author (Professor Matjaz Perc):
Comments to the Author:

AE.1. The two referees rated your work rather differently. We would be happy to consider a resubmission if you can make a strong case for the added value of your work to existing literature. Referee #2 expressed concerns in this regard.

R.AE.1. Thank you very much for the opportunity to revise the manuscript. I have considered all comments carefully, paying special attention to the note of Referee #2 on the novelty of the contents (see R.2.1. below). In addition, several small changes were made to address the comments globally.

Reviewer comments to Author:

Reviewer: 1

Comments to the Author(s)

1.1. In "Assessing the strength of case growth trends in the coronavirus pandemic" authors study a critically relevant and timely problem, and they provide useful insights into how to better understand the spread of COVID-19 infections based on assessing the strength of case growth trends.

R.1.1. Thank you for reviewing the manuscript.

1.2. I have enjoyed reading, and I am in favor of publication after suitable revisions in the Royal Society Open Science.

R.1.2. Thank you for providing clear suggestions for the revision of the contents.

1.3. A relevant recent reference to be added for a more detailed coverage of similar approaches is Forecasting COVID-19, Front. Phys. 8, 127 (2020) and Early spread of COVID-19 in Romania: Imported cases from Italy and human-to-human transmission networks, R. Soc. Open Sci. 7, 200780 (2020).

R.1.3. I have incorporated these studies into the revised manuscript (Background, p. 3, line 35-40).

1.4. It would also improve the paper if the figure captions would be made more self contained. In addition to what is shown for which parameter values, one could also consider a sentence or two saying what is the main message of each figure.

R.1.4. I have added the main message of each figure to the figure captions.

1.5. Finally, it would be useful if the authors would make their source code available as supplementary material. This would promote the usage of the proposed method and allow others to take advantage of this research, and also allow them to reproduce the results.

R.1.5. As for the original submission, I was happy to provide the program code and the analyzed data as supplementary materials for the revised manuscript. The used software (R) is completely open

source, so that every single function used for the analysis and displaying the data graphically can be scrutinized, adapted, and reused.

Reviewer: 2

Comments to the Author(s)

2.1. This is a nice exercise using a well-known method with data from the corona-virus pandemic.

R.2.1. Thank you very much reviewing the manuscript. I think I did not manage to make clear in the original submission that the proposed indicators are far from being well-known. On the opposite, according to my knowledge, they have not been considered as candidates for epidemic surveillance at all yet. In my opinion, it is not unlikely that they are completely new to most biologists, ecologists, and epidemiologists. I have made this issue more clear in the revised manuscript (Background, p. 3, line 44-47; please see also R.2.3).

2.2. The selection of the time and the countries that were chosen is not clear: why India, Israel, Germany and the US? What is special about this countries? The time frame is too short until August.

R.2.2. Both the timeframe and the countries were selected in order to be able to illustrate the main characteristics of the proposed measures. The first SARS-CoV-2 infection was reported in January or February in most countries, therefore, the end date for the analysis (August 4, 2020) marks approximately the initial half-year of the pandemic. Countries differed substantially regarding the development of their SARS-CoV-2 case counts during this first half year, providing an excellent opportunity to describe the behaviour of the proposed indicators for various conditions. On the other hand, completely new patterns have not emerged after August 2020, making an extension of the observation period unnecessary, or even unnecessarily complicating the main message. The countries that were chosen for illustration showed particularly clear patterns and could be used as prototypical examples; no other reason was considered beyond that. Please notice that the analyses have been performed for several other countries as well, reported in the supplementary materials. I have described his rationale more clearly in the revised manuscript (Methods, p. 3, line 58-66; Results, p. 5, line 138-142).

2.3. I am not convinced that moving average is adding substantially information to the many other models that have been published recently.

R.2.3. Apparently I did not manage to highlight the innovative aspects of the study comprehensibly in the original submission. I apologize for that. I agree that moving averages are well known and do not add much to existing knowledge. However, according to the knowledge of the author, the two other measures (the permutation entropy and the relative strength index) have not been yet considered as possible measures of disease surveillance (apart from one study using permutation entropy to quantify predictability, cited in the manuscript). They are very likely to be new to most computational biologists and infectious disease epidemiologists both regarding their conceptual focus (trend strength rather than direction or magnitude) and their application. I have emphasized this point more strongly in the revised manuscript (Background, p. 3, line 44-47).